# Enhancing Therapeutic Approaches for Melanoma Patients Targeting Epigenetic Modifiers

**DOI:** 10.3390/cancers13246180

**Published:** 2021-12-08

**Authors:** Maria Gracia-Hernandez, Zuleima Munoz, Alejandro Villagra

**Affiliations:** Department of Biochemistry and Molecular Medicine, School of Medicine and Health Sciences, The George Washington University, Washington, DC 20052, USA; mariaghernandez@gwu.edu (M.G.-H.); zmunoz09@gwmail.gwu.edu (Z.M.)

**Keywords:** epigenetics, melanoma, acetylation, methylation, combination therapy, resistance, targeted therapy, immunotherapy, adjuvant

## Abstract

**Simple Summary:**

Melanoma affects over 300,000 people worldwide every year. Recent advancements in therapeutic treatments for melanoma patients, such as targeted therapies and immunotherapy, have improved the survival of patients without advanced disease. However, an important subset of patients remains refractory or develops resistance. Melanomagenesis, disease progression, and resistance to therapies are epigenetically regulated processes. Emerging preclinical and clinical research elucidates the mechanisms by which epigenetic drugs can prevent resistance or enhance the therapeutic efficacy of the aforementioned therapies in addition to chemotherapy, radiation therapy, and others. In this review, we assess the role of epigenetics in melanoma progression and resistance to targeted and immune therapies. Additionally, we discuss recent preclinical and clinical reports evaluating the use of epigenetic drugs as adjuvants to enhance the current therapeutic approaches for melanoma patients.

**Abstract:**

Melanoma is the least common but deadliest type of skin cancer. Melanomagenesis is driven by a series of mutations and epigenetic alterations in oncogenes and tumor suppressor genes that allow melanomas to grow, evolve, and metastasize. Epigenetic alterations can also lead to immune evasion and development of resistance to therapies. Although the standard of care for melanoma patients includes surgery, targeted therapies, and immune checkpoint blockade, other therapeutic approaches like radiation therapy, chemotherapy, and immune cell-based therapies are used for patients with advanced disease or unresponsive to the conventional first-line therapies. Targeted therapies such as the use of BRAF and MEK inhibitors and immune checkpoint inhibitors such as anti-PD-1 and anti-CTLA4 only improve the survival of a small subset of patients. Thus, there is an urgent need to identify alternative standalone or combinatorial therapies. Epigenetic modifiers have gained attention as therapeutic targets as they modulate multiple cellular and immune-related processes. Due to melanoma’s susceptibility to extrinsic factors and reversible nature, epigenetic drugs are investigated as a therapeutic avenue and as adjuvants for targeted therapies and immune checkpoint inhibitors, as they can sensitize and/or reverse resistance to these therapies, thus enhancing their therapeutic efficacy. This review gives an overview of the role of epigenetic changes in melanoma progression and resistance. In addition, we evaluate the latest advances in preclinical and clinical research studying combinatorial therapies and discuss the use of epigenetic drugs such as HDAC and DNMT inhibitors as potential adjuvants for melanoma patients.

## 1. Introduction

Melanoma is the least common but deadliest type of skin cancer. In 2020, approximately 325,000 new melanoma cases were identified worldwide, and over 57,000 deaths occurred in the same year [1]. It is estimated that in the United States alone, more than 100,000 new cases will be diagnosed in 2021, and that approximately 7000 patients will succumb to the disease [2]. The current standard of care for melanoma patients includes surgery, targeted therapy, and immunotherapy, along with other therapeutic options such as chemotherapy, radiation therapy, oncolytic viral therapy, among others. However, surgical removal or excision has been the only option for melanoma patients for decades, and the lack of targeted therapies, extensive sun exposure, and tanning beds in developed countries have led to increased incidence and mortality rates [3,4].

The discovery of new treatments for patients with advanced disease and raising awareness of early melanoma signs have resulted in a recent decline in mortality [2,4]. The U.S. Food and Drug Administration (FDA) approved over ten novel therapies for melanoma patients during the past decade, including MAPK inhibitors and immune checkpoint blockade (ICB). When all stages are combined, these approvals have resulted in a 5-year survival rate of 93% [2], although it is well-known that patients with advanced disease have a significantly lower survival rate. Moreover, an important subset of melanoma patients remains unresponsive to these therapies or develops resistance over time; 15–20% of melanoma tumors harboring the BRAF^V600E^ mutation do not respond to targeted therapies such as BRAF and MEK inhibitors (BRAFi, MEKi, respectively) [5], and treatment with ICB such as anti-PD-1 (programmed cell death protein 1) or anti-CTLA4 (cytotoxic T-lymphocyte–associated antigen 4) does not confer a therapeutic benefit to 40–60% of patients [6]. 

To overcome these limitations, several combinatorial therapies have been proposed [7,8,9]. Some of these new modalities, such as the combination of ICB and BRAFi, have been tested. Initial clinical results have shown improvements limited to a subset of patients, with a large proportion of patients being refractory or becoming resistant to these therapies [10]. Therefore, it is critical to identify new combination therapies that can potentiate the therapeutic effect of ICB and targeted therapies or that can overcome or reverse resistance. 

Epigenetics refers to changes in gene expression without altering the DNA sequence, thus resulting in heritable phenotypic modifications [11,12]. Some epigenetic modifications include acetylation, deacetylation, and methylation of histones by histone acetyltransferases (HATs), histone deacetylases (HDACs), and histone methyltransferases (HMTs). Although HATs, HDACs, and other epigenetic modifiers control the epigenome by acting over chromatin, some epigenetic modifiers like HDACs can also modify non-histone proteins and influence key cellular regulatory processes [13,14]. Epigenetically-regulated cellular processes include proliferation, drug sensitivity, resistance, and immune-related functions. Therefore, abnormal epigenetic alterations lead to the development and progression of several cancers and to the resistance to multiple treatments [15]. As a result, the use of epigenetic drugs as adjuvants is being extensively studied to potentiate the therapeutic efficacy of immunotherapy and targeted therapies and to prevent epigenetically driven resistance mechanisms [16,17,18]. 

This review evaluates the role of epigenetic alterations in melanoma progression and resistance to targeted and immune therapies. We also review the latest advances in preclinical and clinical research regarding therapies for melanoma patients. Finally, we assess the use of epigenetic modifiers in combinatorial therapies and adjuvants for melanoma patients.

## 2. Role of Epigenetic Modifiers in the Regulation of Tumor Suppressor Genes and Oncogenes in Melanoma Progression

Melanoma originates from the malignant transformation of melanocytes that have undergone multiple genetic and epigenetic alterations [16]. Melanoma is thought to arise in a stepwise manner that allows for its progression into metastatic disease [19]. It starts as a benign nevus formed by melanocytes that have proliferated aberrantly and becomes a hyperplastic senescent lesion [19]. Once senescence is overcome, the nevus becomes dysplastic and progresses into a radial growth phase confined to the epidermis that later invades the dermis during the vertical growth phase, ultimately leading to metastasis [19]. This stepwise process is driven by the dysregulation of both tumor suppressor genes (TSGs) and oncogenes [19,20,21]. 

Although one of the best-known oncogenic drivers in melanoma is the BRAF^V600E^ mutation, which is found in 66% of malignant melanomas [22], it is not epigenetically regulated during melanoma progression. However, this point mutation is essential for melanoma development in the nevi, which is a critical step for initiating melanocytic neoplasia [19,21,23,24]. Despite the importance of the BRAF^V600E^ mutation for melanomagenesis, additional passenger mutations are required for disease progression. Therefore, melanoma progression depends on other alterations that include but are not limited to chromosomal deletions, inactivating mutations of TSGs, activating mutations in oncogenes, and epigenetic alterations thereof [21,25]. In this review, we discuss the therapeutic effects of targeting epigenetic modifiers in the context of chromatin regulation and non-chromatin scenarios.

To illustrate the importance of epigenetic alterations for melanoma progression, it has been reported that melanoma metastases have a higher expression of H3K27me3 compared to primary melanomas [26]. Epigenetic dysregulation of SWI/SNF chromatin remodeling complexes and TSGs such as *CDKN2A*, MTAP, PTEN, RASSF1A, APAF-1, and P53 contribute to melanoma progression and invasion [19,23,27]. Additionally, oncogenes that can be regulated by epigenetic modifiers and support melanoma progression and development include RAS, ERK, c-jun, MITF, MDM2, and BCL-2 [19,23]. In this section, we provide an overview (Table 1) of how epigenetic modifiers regulate TSGs and oncogenes that are critical for melanoma formation and progression. 

### 2.1. Tumor Suppressor Genes

TSGs are genes involved in the control of cell cycle, growth, and proliferation. Because of their vital role in these processes, they are frequently silenced in cancer by different mechanisms such as promoter hypermethylation and inactivating mutations. 

The *CDKN2A* (cyclin-dependent kinase inhibitor 2A) gene locus is located in the 9p21 chromosomal region and encodes for p16INK4A and p14ARF, which control the G1-S checkpoint in the cell cycle. *CDKN2A* is a melanoma susceptibility gene mutated in 20–50% of familial melanoma cases and 2–3% of sporadic melanomas [28]. *CDKN2A* promoter hypermethylation occurs in uveal melanoma [29] and cutaneous melanoma [28]; p14ARF expression is inversely correlated with melanoma progression since its expression decreases in the progression from melanocytic nevi to the metastatic state [49]. Approximately 20% of primary melanoma tumors in the vertical growth phase showed increased promoter hypermethylation, which was associated with increased proliferation and decreased patient survival [28,29,50]. Promoter hypermethylation is more common in primary melanoma samples from the vertical growth phase as compared to the radial growth phase, and it is associated with histone methyltransferase SETDB1 expression [28]. Aberrant methylation of this locus provides an advantage for melanomas to grow and metastasize, thus highlighting the role of epigenetic alterations in melanoma progression.

Located within the same chromosomal region is *MTAP* (methylthioadenosine phosphorylase), a critical TSG in melanoma. The *MTAP* gene promoter is hypermethylated in eight out of nine human melanoma cell lines [30]. This epigenetic change is essential because MTAP expression negatively correlates with tumor progression in vivo as observed in the consecutive decrease in expression as the disease progresses [30,31]. MTAP deletions increase methylthioadenosine (metabolite cleaved by MTAP), which inhibits PRMT5 (protein arginine methyltransferase 5) [51,52], thus leading to hypomethylation of genomic regions controlled by PRMT5. Therefore, MTAP is not only dysregulated in melanoma by promoter hypermethylation, but its loss can also impair epigenetic mechanisms mediated by PRMT5. This example highlights the cascade effect that epigenetic aberrations can have on other cellular pathways, including further epigenetic alterations. 

PTEN (phosphatase and tensin homolog) is a phosphatase that converts PIP3 (phosphatidylinositol (3,4,5)-phosphate) to PIP2 (phosphatidylinositol 4,5-bisphosphate), thus suppressing the activation of the PI3K/Akt signaling pathway. PTEN is frequently mutated across different types of cancer, including melanoma, with a reported rate in melanoma cell lines of 30–50% and 5–20% in primary samples [53]. However, epigenetic silencing of PTEN is more frequent than point mutations in melanoma as PTEN promoter methylation occurs in 62% of samples from melanoma patients [32]. Another study using cutaneous melanoma samples from 230 patients found the PTEN promoter to be methylated in 60% of the samples, which was associated with poor prognosis [33]. PTEN can also be regulated by an epigenetic mechanism of biallelic functional inactivation in malignant melanoma tumors without PTEN mutations [34]. Besides promoter hypermethylation, PTEN function can also be regulated at the protein level by epigenetic modifiers. Specifically, PTEN acetylation at Lys163 increases its translocation to the plasma membrane and helps downregulate the PI3K/Akt survival pathway [35]. HDAC6 deacetylates PTEN to decrease its phosphatase activity, which can be restored using HDAC6 inhibitors [35]. This example highlights the multiple mechanisms by which epigenetic regulation can lead to the survival of melanoma cells and disease progression. 

*RASSF1A* is a TSG that encodes a microtubule-associated protein involved in processes controlling cell cycle, mitotic arrest, and apoptosis. *RASSF1A* is epigenetically silenced in melanoma through promoter hypermethylation [36,37]. Two regions of the RASSF1A CpG island have been reported to be methylated in multiple metastatic melanoma tumors and cell lines [36]. Interestingly, methylation of the RASSF1A CpG island may be correlated to melanoma development and progression as it has been reported that RASSF1A is less frequently methylated in early-stage melanoma but in 50% of stage III and IV melanoma samples [37], thus suggesting that this epigenetic change contributes to melanoma progression.

Other significant TSGs dysregulated in melanoma include Apaf-1 and p53. Apaf-1 is a cell death effector that mediates p53-dependent apoptosis, and p53 is a well-studied TSG that controls cell cycle arrest and apoptosis. Although p53 is rarely mutated in malignant melanoma as compared to other types of cancer [54,55], other proteins that are important for p53-dependent apoptosis, such as Apaf-1, can be mutated or epigenetically silenced. Apaf-1 is epigenetically silenced in metastatic melanoma by promoter hypermethylation and loss of heterozygosity [38]. Apaf-1 is regulated at the chromatin level, whereas p53 function can be regulated through acetylation at the protein level. HDAC6 deacetylates p53 at Lys381/382, which coordinates p53-induced apoptosis [39]. It has been reported that the selective HDAC6 inhibitor A452 can modulate p53 expression levels differently depending on its mutational status. A452 increases wild-type p53 levels by destabilizing MDM2 and decreases mutant p53 by inducing MDM2 [39]. In summary, p53-dependent apoptosis is regulated at the chromatin and protein levels, thus demonstrating that epigenetic modifiers can modulate multiple processes in a chromatin scenario like changes in gene expression and post-translational modifications of proteins that participate in key cellular pathways.

### 2.2. Oncogenes

Oncogenes promote cell growth and division and, when mutated, lead to uncontrolled cell proliferation. Genes that promote cell division can be overexpressed or overactivated in cancer by different mechanisms, such as activating mutations and copy number alterations. However, these oncogenes can also be regulated by epigenetic modifiers. 

The MAPK signaling pathway is one of the most frequently mutated pathways in cancer as it promotes cell division. This section of the review focuses on three members of this pathway that can be regulated by epigenetic modifiers: RAS, ERK, and c-jun. RAS is mutated in many types of cancer as it is a crucial activator of this pathway. K-RAS can be acetylated at Lys104 to attenuate its transforming activity, while mutant K-RAS has been reported to be deacetylated by HDAC6 and SIRT2 [40]. Therefore, when mutant K-RAS tumors are treated with HDAC6 or SIRT2 inhibitors, cell growth decreases [40]. Downstream signaling of RAS also has epigenetic consequences at the chromatin level. Specifically, oncogenic N-RAS increases global H3K9ac/H3K23ac levels at the Egr1 and JunB promoters [56], thus modulating H3 acetylation and further regulating the expression of other genes. 

Similarly, post-translational acetylation and deacetylation of ERK1/2 modulate its activity. ERK1/2 can be acetylated by the acetyltransferases CREB-binding protein (CBP) and p300 and deacetylated by HDAC6 [41]. This acetylation occurs at Lys72, which affects its ATP-binding domain, and thus modulates its activity [41]. Specifically, HDAC6 stimulates ERK1/2 activity by deacetylating it at Lys72 [41]. HDAC6 not only deacetylates RAS and ERK1/2 in the MAPK pathway, but also c-jun, which is acetylated by p300/CBP at Lys271 [42]. HDAC6 can deacetylate c-jun NH2-terminal kinase-mediated beclin 1 and induce autophagic cell death [57]. In addition, c-jun can also be deacetylated by HDAC8, which increases its transcriptional activity [43]. 

Besides the MAPK pathway, MITF is very important for melanoma progression as it controls melanocyte-specific proliferation [44]. Interestingly, MITF is one of the most significantly upregulated genes under the control of the BRAF^V600E^ mutation [58]. As a transcription factor, MITF recruits epigenetic machinery to melanocyte-specific promoters to induce their expression. In particular, MITF requires the chromatin-remodeling complex SWI/SNF to activate melanocyte-specific genes [59]. The transcriptional activation of these genes is partly mediated by the acetyltransferase CBP/p300 [46], and the interaction between CBP/p300 and MITF is dependent on MITF phosphorylation by ERK2 at Ser73 [60]. Additionally, the methylation patterns of the MITF promoter can change during melanoma progression, and it can be reactivated in melanoma by hypomethylation [45].

Other oncogenes that are epigenetically regulated and important for melanoma tumorigenesis include MDM2 and BCL-2. MDM2 is a negative regulator of p53, while BCL-2 is an antiapoptotic protein. MDM2 recruitment of a protein complex containing HDAC1 which deacetylates p53 at all the acetylation sites known and promotes its degradation negatively regulates p300/CBP-mediated p53 acetylation [47]. BCL-2 is overexpressed in both primary and metastatic melanoma lesions, which is associated with disease progression [61]. CBP and HDAC2 regulate BCL-2 expression, and treatment with HDAC inhibitors can reduce their binding as reported in a lymphoma model [48]. Furthermore, bcl-xL expression increases as melanomas evolve from primary to metastatic lesions, thus indicating that bcl-xL increases melanoma’s malignant potential [61].

In summary, TSGs and oncogenes that are essential for melanoma tumorigenesis and progression to metastatic disease can be regulated at both the chromatin and protein levels by epigenetic modifiers. These regulatory events are not isolated and require other mutations, genetic alterations, or dysregulation of epigenetic mechanisms for the establishment and progression of the disease. Nevertheless, it is critical to consider the high degree of variability of the tumor models used to investigate these phenomena as epigenetic alterations may be different between cell lines routinely used in research laboratories, freshly isolated tumors from in vivo studies, or primary tumors from cancer patients [62]. Despite this variability, epigenetic machinery represents a potential therapeutic target either as a standalone therapy or as adjuvants to improve the current therapeutic approaches available for melanoma patients.

## 3. Role of Epigenetics in Resistance to Targeted Therapies and Immunotherapy

The discovery of targeted and immune therapies was a major breakthrough in the treatment of melanoma patients. Although some patients respond to these therapies, their therapeutic benefit is limited by the existence of primary resistance in the tumors and the emergence of resistance during treatment. Among the multiple mechanisms of resistance, epigenetic alterations play a critical role in developing resistance. Resistance to therapies can be intrinsic if mediated by the tumor cells or extrinsic if conferred by other cells in the tumor microenvironment, such as immune cells, fibroblasts, or stromal cells [63]. This section focuses on specific epigenetic alterations that contribute to intrinsic and extrinsic mechanisms of resistance to MAPK inhibitors and ICB.

### 3.1. Resistance to BRAF and MEK Inhibitors

It is estimated that about half of melanoma patients have mutations in BRAF, with BRAF^V600E^ representing approximately 80% of these mutations [64]. Therefore, determining BRAF mutational status in melanoma patients helps clinicians determine the first line of treatment. As such, BRAFi such as vemurafenib, dabrafenib, or encorafenib represent the standard of care for many melanoma patients [65]. Although these targeted therapies represented a breakthrough in terms of treatments for melanoma patients and resulted in improved survival, the emergence of resistance limits the therapeutic efficacy of these therapies. Intermittent treatment schedules, or drug holidays, represent a strategy to delay resistance to BRAFi [66], suggesting that some resistance mechanisms might be reversible and thus epigenetically regulated. Therefore, combining BRAFi with MEKi such as trametinib, cobimetinib, or binimetinib, represents the current approach [65]. 

Resistance to BRAFi and MEKi, collectively known as MAPK inhibitors, arises from genetic and epigenetic alterations [67]. As reviewed by Proietti et al., genetic mechanisms that lead to resistance to targeted therapies include mutations in RAS, BRAF, and MEK, differential splicing of BRAF, BRAF amplification, among others [67]. Besides genetic alterations, epigenetic disturbances also play an important role in resistance to targeted therapies. 

After the drug-sensitive cells succumb to treatment, a small subset of quiescent cancer cells has the potential to survive targeted therapy and become resistant by undergoing cell reprogramming that can lead to the activation of survival pathways [68,69]. These changes allow resistant cells to evolve and resist therapy by altering their epigenome. A genome-wide CRISPR/Cas9 screening of melanoma cells revealed that mutations in several genes related to histone modifications contribute to resistance to vemurafenib [70]. Specifically, researchers found mutations in multiple genes that form the STAGA complex, a chromatin-acetylating transcription coactivator that regulates mRNA splicing, transcription, and DNA damage, and in SMARCA4, an SWI/SNF-related protein [71]. Additionally, the altered chromatin state that supports resistance to these therapies is also characterized by increased expression of histone demethylases such as KDM5A and KDM5B, thus contributing to melanoma progression [72,73,74]. Moreover, these epigenetic alterations increase the expression of drug efflux genes, melanoma stem cell markers, and other histone-modifying enzymes [75].

It was recently reported that upregulation of multiple HDACs contributes to resistance. HDAC6 induces resistance to vemurafenib by preventing vemurafenib-induced cell death [76]. HDAC8 is also involved in therapeutic resistance to BRAFis, mediated by increased c-jun transcriptional activity [43]. Thus, treatment with specific HDAC6 and HDAC8 inhibitors could enhance the therapeutic efficacy of BRAFi. Additionally, nonspecific pan-HDAC inhibitors such as panobinostat could also be used for these purposes as they can restore sensitivity to BRAF inhibition in resistant cells showing partial response [77]. 

### 3.2. Resistance to Immunotherapy

Immunotherapy aims to help the patient’s immune system fight the disease. Particularly for melanoma, the most common immunotherapy approach used comprises immune checkpoint blockade (ICB). These antibodies target negative regulators of T cell activation to prevent them from being activated, thus resulting in prolonged T cell activation in the tumor and enhanced antitumor response. About 50% of melanoma patients with advanced or metastatic disease have durable responses to ICB [78,79]. Despite the success that multiple immunotherapeutic agents have had on the survival of melanoma patients, there is an important subset of patients who develop resistance or whose disease progresses after these therapies. Although multiple intrinsic and extrinsic mechanisms of resistance to immunotherapies have been found to date in melanoma, those of epigenetic origin are yet to be fully elucidated. 

To date, differential DNA methylation patterns have been identified between melanoma patients that were treated with anti-CTLA4 and that had a clinical benefit compared to those with no clinical benefit [80]. The differentially methylated genes were involved in nervous system development, differentiation, and function. Interestingly, researchers have found that patients that respond to anti-PD-1 therapy have increased neural crest markers and decreased melanocytic markers [81]. This suggested dedifferentiation of melanoma cells upon ICB is driven by intrinsic epigenetic changes that lead to hyper-accessible chromatin regions. Other intrinsic mechanisms of resistance include overexpression of PD-L1 which is the ligand for PD-1. Since HDAC6 regulates PD-L1 expression in melanoma cells through STAT3 signaling, HDAC6 inhibitors can be used to decrease PD-L1 expression, thus reversing this resistance [82]. Moreover, the histone demethylase LSD1 contributes to resistance to immunotherapy, and inhibition of LSD1 can improve the antitumor effect of anti-PD-1 by enhancing the immunogenicity of melanoma cells and increasing T cell infiltration [83]. 

Besides, EZH2 is important for melanoma progression as aberrations are found in 27% of patients, and it is associated with invasive melanoma [84,85]. This histone methyltransferase represses the expression of genes that contribute to the antitumor immune response, such as RASSF5 and ITGB2, which positively correlate with infiltration of dendritic cells (DCs) and CD4 and CD8 T cells in tumors [86]. Additionally, T cell accumulation in the tumor during anti-CTLA-4 or IL-2 immunotherapy leads to an increase in EZH2 expression in melanoma cells, which further decreases their immunogenicity and antigen presentation, thus contributing to intrinsic resistance to these immunotherapies [87]. Because aberrations in EZH2 can hinder the T cell-mediated effects of ICB, EZH2 inhibitors represent a potential adjuvant to enhance the antitumor immune response elicited by ICB [88]. Lastly, melanoma is resistant to anti-CD47 blockade, a novel immunotherapy approach that aims to enhance macrophage phagocytosis of cancer cells, via an evolutionarily conserved mechanism [89] that could be epigenetically regulated. 

As previously mentioned, extrinsic epigenetic mechanisms of resistance include those coming from the immune cells, stromal cells, and fibroblasts found in the tumor microenvironment [63]. Despite the importance of some immune cells such as T cells for antitumor responses, some immune cells like macrophages contribute to tumor growth. Specifically, protumoral or alternatively activated macrophages (M2) and tumor-associated macrophages (TAMs) contribute to resistance to immunotherapy through extrinsic mechanisms [90]. Specifically, M2 macrophages and TAMs limit the efficacy of ICB by preventing CD8 T cells from reaching the tumor core and by preventing phagocytosis through PD-1 expression [91,92]. Thus, macrophages are interesting targets for cancer immunotherapy and other diseases because of their phenotypic plasticity and variety of functions [90,93]. Therefore, the modulation of macrophage phenotype by epigenetic modifiers is an active research area. HDAC6 inhibitors decrease the expression of the immunosuppressive cytokine IL-10 in macrophages [94,95,96,97] and decrease the M2 phenotype [98]. Additionally, low doses of nonspecific HDAC inhibitors such as trichostatin A convert TAMs into antitumoral, proinflammatory macrophages [99].

In addition, researchers have found an inverse correlation between 4-1BB DNA methylation and mRNA expression with disease progression, where 4-1BB hypermethylation correlates with poor response to anti-PD-1 [100]. Additionally, 4-1BB hypomethylation correlates with higher overall survival and increased immune infiltration in the tumor microenvironment, thus representing a potential biomarker for response to anti-PD-1 [100]. 4-1BB or CD137 is a costimulatory molecule found in different immune cells such as T cells, natural killer (NK) cells, activated DCs, monocytes, B cells, and neutrophils [101]. 

The extrinsic and intrinsic epigenetic mechanisms by which melanoma patients develop resistance to targeted and immune therapies are still being investigated. However, preclinical and early clinical data currently available suggest that the combination of epigenetic modifiers with other therapies can have advantageous effects in the clinic, such as overcoming of resistance to first-line therapies. Thus, epigenetic modifiers can serve as adjuvants to potentiate the clinical benefit of the current therapies for melanoma patients.

## 4. Epigenetic Modifiers as Therapeutic Adjuvants for Melanoma Patients

Epigenetic drugs are attractive therapeutic agents for the treatment of multiple cancer types. Some small-molecule inhibitors targeting epigenetic modifiers have been approved by FDA to treat hematological cancers, but they have not proven effective for solid cancers [102]. Although pan-HDAC inhibitors such as panobinostat exert tumor cytotoxicity and increase immunogenicity in mouse melanoma models [103], the therapeutic potential of this drug in the clinic is limited by the toxicity it induces in patients [104]. 

Although selective HDAC inhibitors induce less cytotoxicity and modulate antitumor immunity and cell proliferation [105], they do not work well as standalone therapy. For this reason, combinations of different epigenetic drugs such as DNA methylation and HDAC inhibitors are being investigated and show positive results [106]. Additionally, the combination of HDAC inhibitors and DNMT inhibitors may restore the expression of silenced genes such as TSGs in the cancer cells, in addition to their immunomodulatory effects [107]. Besides DNMT inhibitors, HDAC inhibitors are also being investigated in combination with other epigenetic drugs such as bromodomain and extraterminal motif (BET) inhibitors as they synergize to induce apoptosis of melanoma cells and reduce tumor growth in mouse models [108]. The combination of multiple selective HDAC inhibitors acts synergistically in T cell lymphoma cell lines [109]. This suggests that, despite the cytotoxic effects of pan-HDAC inhibitors, the combination of multiple highly selective HDAC inhibitors may prove successful. However, it is critical to evaluate specific combinations of highly selective epigenetic drugs in order to elucidate the mechanisms by which combinatorial epigenetic therapies have the highest immunomodulatory and therapeutic effects in patients without increasing the potential toxicity that may arise from the combination.

The multiple cellular pathways that are epigenetically regulated in both cancer and immune cells make epigenetic modifiers a potential target to simultaneously modulate cellular processes that regulate both intrinsic and extrinsic mechanisms of resistance. Additionally, epigenetic drugs can sensitize solid tumors to cytotoxic drugs that are FDA approved and used in the clinic. Encouraging results have increased researchers’ interest in studying the safety and therapeutic potential of epigenetic drugs for the treatment of melanoma in preclinical research and in clinical trials, particularly in combination with standard-of-care therapeutic agents such as targeted therapies and ICB and other treatment avenues like chemotherapy, radiation therapy, nanoparticle-based therapies, or other immune cell-based therapies [110]. This section gives an overview (Figure 1) of the latest research that evaluates the combination of HDAC inhibitors or methylating agents with the aforementioned therapies in addition to chemotherapy, radiation therapy, nanoparticles, and others. A summary table of the epigenetic drugs discussed in this review article can be found in Table 2.

### 4.1. Targeted Therapies (MAPK Inhibitors)

As previously mentioned, BRAF^V600E^ mutations have been reported in approximately 50% of melanoma patients and represent the most common BRAF mutation, which activates the MAPK pathway [22]. In most cases, patients with BRAF mutant metastatic melanoma are treated with BRAFi alone or combined with other MAPK inhibitors such as MEKi. However, most patients develop resistance to treatment over time, partially due to the emergence of secondary mutations [111]. Multiple studies have shown that combining BRAFi with epigenetic drugs such as HDAC inhibitors can sensitize melanoma cells to MAPK inhibitor-induced cell death and reverse resistance to these therapies by eliminating resistant clones. 

Multiple research groups have found similar results regarding the potential of combining epigenetic drugs with MAPK inhibitors to sensitize cells to these therapies. Combining HDAC inhibitors such as panobinostat or suberoylanilide hydroxamic acid (SAHA) with the BRAFi PLX4720 or vemurafenib can induce melanoma cell death by activation of noncanonical cell death pathways [112]. Selective HDAC inhibitors can also be used for these purposes. The HDAC6 inhibitor ACY-1215 sensitized melanoma cells to vemurafenib by inactivating ERK and promoting endoplasmic reticulum stress [76]. 

Moreover, this combination can eliminate BRAF inhibitor-resistant melanoma cells. Treatment with the HDAC inhibitor vorinostat depleted MAPK inhibitor-resistant cells by increasing reactive oxygen species, increasing drug-sensitive clones that succumbed to BRAFi [113]. Another study showed that long-term treatment with the BRAFi PXL4032 on sensitive BRAF mutant melanoma cell lines induced senescent resistant clones that portrayed a stem cell-like phenotype and had an aberrant expression of multiple epigenetic modifiers such as HDAC6 [114]. This senescent state was reversed by HDAC inhibitors such as SAHA and mocetinostat (MGCD0103) [114].

The synergistic effects of these combinations are not only applicable to cutaneous melanoma but also uveal melanoma. Researchers have found that uveal melanoma patients can develop resistance to MEKi by upregulating the PI3K/Akt pathway [115]. This upregulation serves as an escape mechanism, which epigenetic drugs can limit. Among them, the HDAC inhibitor panobinostat was more effective at inhibiting the adaptive PI3K/Akt pathway and YAP signaling than the DNMT inhibitor decitabine, the HAT inhibitor anacardic acid, or other HDAC inhibitors like the HDAC1/2/3 inhibitor entinostat, the HDAC6 inhibitor tubastatin, or the HDAC8 inhibitor PCI-34051 [115].

### 4.2. Immune Checkpoint Blockade

Immune checkpoint blockade (ICB) represents one of the major breakthrough treatments for melanoma patients. However, only 40 to 60% of melanoma patients respond to ICB, with much lower response rates in metastatic melanoma patients [78]. ICB is used to prevent T cell inactivation, and the most common antibodies used to treat melanoma patients include nivolumab and pembrolizumab, which prevent PD-1 from interacting with PD-L1, and ipilimumab, which blocks CTLA-4.

Treatment-related adverse events are reported more frequently in ipilimumab-treated patients compared to nivolumab-treated patients [79,116]. However, a clinical trial demonstrated that the combination of guadecitabine and ipilimumab was safe in patients with unresectable stage III/IV melanoma and that it upregulates HLA class I in melanoma cells and enhances antitumor immunity as demonstrated by an increase in CD8+ PD-1+ T cells and CD20+ B cells [117]. Preclinical research has demonstrated that EZH2 inhibition synergizes with anti-CTLA-4 in mouse melanoma models, where EZH2 inhibition restored melanoma immunogenicity and antigen presentation [87].

Preclinical research evaluating the combination of epigenetic modifiers with ICB is emerging. It has been reported that pretreatment with pan-HDAC inhibitors such as AR42 and sodium valproate enhances the therapeutic efficacy of anti-PD-1 and anti-CTLA-4 antibodies in B16 mouse melanoma models [118]. Treatment with HDAC inhibitor and anti-PD-1 led to increased T cell, antitumoral macrophage, neutrophil, and NK cell infiltration in the tumors [118,119]. 

Recently, our group has demonstrated that pretreatment with selective HDAC6 inhibitors such as nexturastat A and suprastat before treatment with anti-PD-1-blocking antibodies enhances the therapeutic efficacy of anti-PD-1 checkpoint blockade in melanoma and breast cancer mouse models by downregulating immunosuppressive molecules, decreasing protumoral M2 macrophages, and increasing T cell infiltration [98,120,121]. In addition, treating T cells isolated from metastatic melanoma patients with the HDAC6 inhibitors ACY-1215 and ACY-241 decreases Th2-associated cytokines, increases Th1-associated cytokines, decreases FOXP3 expression and T cell exhaustion markers, and increases the accumulation of central memory T cells in the tumors [122]. However, these results were not obtained with pan-HDAC inhibitors as they exerted cytotoxicity in those T cells [122]. Altogether, these results suggest that selective HDAC inhibitors represent a potential adjuvant to enhance immunotherapy in melanoma patients as they can be used to modulate T cell activity, decrease protumoral macrophages and immunosuppression, and increase the antitumor immune response. In addition, these results also suggest the importance of using highly selective HDAC inhibitors over nonselective inhibitors to modulate immune-related pathways without inducing cytotoxicity.

A clinical study also investigated entinostat’s safety and therapeutic efficacy, a Class I HDAC inhibitor, in combination with pembrolizumab in melanoma patients with disease progression after anti-PD-1 therapy [123]. This combination appeared to be safe in patients and have promising antitumor activity, as shown by an increase in CD8 T cells and a decrease in myeloid-derived suppressor cells [123]. In addition, entinostat increases NK cell-mediated killing of cancer cells [124].

### 4.3. Other Therapeutic Strategies

Other therapeutic strategies to treat melanoma patients include chemotherapy, radiation therapy, nanoparticle-based therapies, and other immune-related therapies. Although the standard of care for melanoma patients involves targeted and immune therapies, the therapeutic approaches mentioned in this section are second-line to treat patients that do not respond or become resistant to first-line therapies. Preclinical and clinical research to evaluate the therapeutic efficacy of epigenetic drugs used in combination with chemotherapeutic agents, radiation, nanoparticles, or other immune-related therapies is ongoing.

#### 4.3.1. Chemotherapy

Chemotherapy is mostly used to treat stage IV melanoma or patients that do not respond to targeted or immune therapies. Common chemotherapy drugs used to treat melanoma patients include alkylating agents such as dacarbazine and temozolomide, although other chemotherapeutic drugs can also be used or are under investigation. Single-agent chemotherapy is well-tolerated, but only 5–20% of patients respond to this therapy [125]. Similarly, combination chemotherapy does not provide additional response rates compared to monotherapy [125].

Preclinical and clinical research is evaluating the combination of single epigenetic modifiers with other chemotherapeutic agents. Nonselective HDAC inhibitors such as valproic acid, trichostatin A, or vorinostat are being investigated. In melanoma cells, valproic acid sensitizes human melanoma cell lines to chemotherapy agents like cisplatin and etoposide [126] or temozolomide [127] by inducing cell cycle arrest and suppressing DNA double-strand break repair. When in combination with trichostatin A, etoposide reestablished p53 activity and induced chemoresistance of melanoma cells [128]. Furthermore, targeting the histone deacetylase SIRT2 sensitized melanoma cells to cisplatin [129]. 

Besides preclinical research, these combination therapies are also under clinical evaluation. For example, vorinostat combined with doxorubicin was evaluated in a phase I clinical trial that included melanoma patients who reported minimal response [130]. Additionally, demethylating agents are also being evaluated; 2′-deoxy-5-azacytidine interacts with cisplatin to induce cytotoxicity in human melanoma cells in a synergistic manner [131]. In murine melanoma cells, adozelesin, another DNA-alkylating agent, synergizes with 5-azacytidine [132]. Moreover, the combination of decitabine and carboplatin slowed cell proliferation and induced apoptosis and senescence [133].

HDAC inhibitors and demethylating agents in combination with chemotherapeutic agents potentiate these chemotherapeutic effects. The combination of HDAC inhibitors and DNMT inhibitors increases chromatin accessibility to cisplatin and doxorubicin, thus potentiating the anticancer effect of these cytotoxic drugs [134]. In addition, the combination of the DNMT inhibitor 5-aza-2′-deoxycytidine and the HDAC inhibitor trichostatin A sensitizes melanoma cells to temozolomide [135]. A phase I clinical trial studying the combination of decitabine and panobinostat with temozolomide reported that this triple therapy is safe in resistant metastatic melanoma patients and that 75% of the patients enrolled in the trial had either stable disease or complete response [136]. Altogether, this preclinical and clinical research demonstrates that epigenetic drugs represent encouraging adjuvants for chemotherapy to treat melanoma patients who are unresponsive to other therapies or have advanced disease.

#### 4.3.2. Radiation Therapy

Radiation therapy is a very effective approach to treat isolated solid tumors in situ, but it is less effective at regulating metastatic cancers. Melanoma is known to be resistant to radiation therapy as these tumors have a high ability to repair radiation-induced DNA damage [137,138]. Therefore, a potential approach to improve the outcome of radiation therapy would be the pharmacological targeting of DNA repair machinery. 

Epigenetic drugs represent a potential candidate to enhance the effects of radiation therapy. A panel of pan-HDAC inhibitors including sodium butyrate, tributyrin, phenylbutyrate, and trichostatin A radiosensizited human melanoma cells by enhancing gamma radiation-induced apoptosis and impairing the repair of damaged DNA [139]. Vorinostat also synergizes with radiation therapy through similar mechanisms [140]. Moreover, HDAC inhibitors such as romidepsin, trichostatin A, valproic acid, and vorinostat have a greater sensitization effect with carbon ion radiation compared to gamma irradiation in a mouse melanoma model, mediated by cell cycle arrest [141]. The radiosensitization potential of valproic acid was also tested and proved effective in drug resistant models [142]. Inhibition of HDAC1, HDAC2, and HDAC3 by a class I HDAC inhibitor can suppress DNA double-strand break repair, thus enhancing the effects of radiation therapy [127]. Besides HDAC inhibitors, demethylating agents such as 5-aza-2′-deoxycytidine also sensitize melanoma cells to gamma irradiation, significantly decreasing viability as compared to each treatment alone [143]. Altogether, these reports suggest that HDAC inhibitors or demethylating agents could reverse the intrinsic resistance of melanoma cells to radiation therapy.

#### 4.3.3. Nanoparticle-Based Therapies

Nanoparticles are used to package and deliver different therapeutic agents for localized interventions. Therefore, cutaneous melanoma has the potential to be treated using nanoparticles thanks to its location on the skin. Nanoparticles are tools to package numerous pharmacological agents, thus their use to package and deliver epigenetic drugs is gaining researchers’ attention. 

Retinoid hydroxamic acid nanoparticles, which combine all-trans retinoic acid and vorinostat, were shown to induce apoptosis, prevent colony formations of melanoma cells, and inhibit the growth of a melanoma tumor xenograft [144]. Additionally, nanoparticles can package and deliver highly specific epigenetic modifiers. As such, the HDAC6 inhibitor nexturastat A was encapsulated in indocyanine green poly (lactic-co-glycolic) acid-based nanoparticles to combine photothermal therapy with epigenetic therapy, which induced the expression of costimulatory molecules in melanoma cells and decreased tumor growth in a mouse melanoma model [145]. Overall, nanoparticles can release epigenetic drugs for prolonged periods, whether they be nonspecific or highly specific inhibitors. Nanoparticles have the potential to circumvent one of the most limiting factors of nonselective epigenetic drugs as their ability to deliver therapeutic drugs to localized areas can prevent the systemic adverse events and cytotoxicity induced by nonselective inhibitors. Therefore, their potential to treat melanoma patients should be exploited and investigated further as cytotoxicity is one of the limiting factors when using pan-HDAC inhibitors in the clinic.

#### 4.3.4. Other Immune-Based Therapies

Other immune-related avenues of treatment for melanoma patients include adoptive cell therapies like chimeric antigen receptor (CAR) T cell therapy or T cell receptor (TCR)-engineered T cell therapy, and oncolytic viral therapy such as talimogene laherparepvec (T-VEC) [146,147,148]. CART T cell or TCR-engineered T cell therapies involve the adoptive transfer of genetically engineered T cells into the patient, whereas T-VEC is a treatment avenue that consists of an oncolytic virus that causes tumor cell lysis and release of tumor-associated antigens. This virus is also engineered to release granulocyte-macrophage colony-stimulating factors to recruit DCs to the tumor, leading to the induction of a tumor-specific immune response [148,149].

Although the use of CAR T cell or TCR therapy to treat melanoma patients is at its infancy and is currently being studied in clinical trials, it is likely that these therapies will be used in combination with other avenues of treatment such as ICB, cytokines, or others [150]. Taking into consideration the important role of epigenetic modifiers in T cells and other immune cells [18], it is highly likely that the therapeutic potential of these adoptive cell therapies and oncolytic viral therapies might be enhanced with the combination of epigenetic drugs such as highly selective HDAC inhibitors. For example, the combination of CAR T cell therapy and HDAC11 inhibitors might be successful as HDAC11 is a negative regulator of T cell effector phenotype and function [151].

## 5. Epigenetic Modifiers in Clinical Trials

Intensive research is ongoing to study the clinical benefit of epigenetic drugs and ICB either as standalone therapies or as combination therapy for melanoma patients. To date, most HDAC inhibitors tested in clinical trials for melanoma patients are pan-HDAC inhibitors, which are known to cause serious side effects in patients. Pan-HDAC inhibitors have been shown to successfully treat hematological malignancies. Vorinostat was approved by the FDA to treat cutaneous T cell lymphoma patients after clinical trials showed a 30% response rate [152]. Other HDAC inhibitors that are FDA-approved as anticancer agents for treating B and T cell lymphoma, myeloid leukemia, or multiple myeloma patients include romidepsin, mocetinostat, and fimepinostat [153].

The therapeutic success of epigenetic drugs is yet to be proved in solid tumors as their cytotoxic and off-target effects hinder their therapeutic efficacy and prevent their study in phase III and IV clinical trials [154]. Some of these side effects include grade 3–4 adverse events such as thrombocytopenia, neutropenia, cardiac toxicity, liver toxicity, venous thromboembolic events, pulmonary embolism, and even death [155]. Thus, it is imperative to study the therapeutic efficacy of highly selective HDAC inhibitors and demethylating agents as they are powerful tools that can modulate signaling pathways that contribute to melanoma progression and, more importantly, prevent resistance to the current therapies used in the clinic.

In Table 3, we summarize the clinical studies that have been or are being carried out to study the efficacy of HDAC and DNMT inhibitors in melanoma patients either as a standalone therapy or combined with other therapeutic approaches. As previously stated, the results from clinical trials where epigenetic drugs are used as a standalone therapy (NCT00185302, NCT01065467, and NCT00121225), especially with nonspecific HDAC inhibitors like vorinostat, have not been very successful, partially due to the serious adverse effects induced in patients [156]. Panobinostat, another pan-HDAC inhibitor, is being tested for the treatment of MITF-amplified melanoma (NCT01065467). A phase II clinical trial evaluating the safety of vorinostat in patients with metastatic or unresectable melanoma (NCT00121225) reported that despite some side effects, vorinostat demonstrated early responses and a high proportion of patients had stable disease [156]. However, as stated above, combining these epigenetic drugs with other treatments may be successful. The combination of panobinostat with the alkylating agent temozolomide and the DNMT inhibitor decitabine has been proven safe in a recent clinical trial (NCT00925132). 

Additionally, a few clinical trials are evaluating the combination of different classes of HDAC inhibitors with ICB. For example, a phase I clinical trial (NCT02935790) evaluated the combination of ACY-241, an HDAC6 inhibitor, with ipilimumab and nivolumab. Another phase I clinical trial (NCT03565406) assessed the combination of mocetinostat, a class I HDAC inhibitor, with ipilimumab and nivolumab. Interestingly, a phase I clinical trial (NCT03903458) evaluates the safety of tinostamustine, a fusion molecule composed of the alkylating agent bendamustine fused to vorinostat. Moreover, in a phase I clinical trial (NCT02032810), the pan-HDAC inhibitor panobinostat was administered at different doses with the same dose of ipilimumab to treat patients with stage III or stage IV melanoma that cannot be removed by surgery. A phase II clinical trial (NCT02697630) evaluated the class I HDAC inhibitor entinostat in combination with pembrolizumab for uveal melanoma patients. Additionally, in a phase II clinical trial (NCT02697630), patients with metastatic uveal melanoma treated with the HDAC inhibitor entinostat combined with the PD-1 inhibitor pembrolizumab experienced tumor regression and durable responses [157]. 

Besides HDAC inhibitors, demethylating agents are being evaluated in clinical trials. As such, the safety of decitabine is being assessed in phase I clinical trials in patients with stage III or IV melanoma (NCT00002980, NCT00030615). A phase II clinical trial is evaluating the safety of azacitidine with pembrolizumab for patients with metastatic melanoma (NCT02816021).

Although preclinical research evaluating the use of epigenetic drugs as adjuvants is emerging as they have become a recent research interest, the results from clinical trials are critical to determining their safety and efficacy at potentiating readily available therapies like MAPK inhibitors or immunotherapy. To date, only a limited number of clinical trials have tested combination therapies. Additionally, most early clinical trials evaluate nonselective HDAC inhibitors, which do not continue into later phases because of the off-target and adverse events previously mentioned. However, considering the important role of epigenetics in the emergence of resistance to both targeted and immune therapies, it is clear that more preclinical and clinical testing will be conducted in the near future to elucidate the most potent combinations that will provide a clinical benefit to a subset of melanoma patients. Nevertheless, highly selective epigenetic drugs are necessary as they are less cytotoxic than their broad-spectrum counterparts.

## 6. Conclusions

Melanoma is a complex malignancy whose progression is driven by multiple mutations and epigenetic alterations that lead to the inactivation of tumor suppressor genes and the activation of oncogenes. These genetic and epigenetic alterations induce cell proliferation, growth, transition to a metastatic state, and resistance to different therapies. This review evaluated how multiple relevant tumor suppressor genes and oncogenes undergo epigenetic changes during melanoma tumorigenesis and progression. In addition, we evaluated the role of extrinsic and intrinsic epigenetic alterations in the emergence of resistance to MAPK inhibitors and ICB. Lastly, we discussed the latest research on combinatorial therapies that include epigenetic drugs.

The design and development of new standalone therapies as well as combinatorial therapies for melanoma patients is an especially important research area. Although extensive research is needed to further elucidate the extrinsic and intrinsic mechanisms by which epigenetic alterations contribute to resistance and can synergize with other therapies, the results from preclinical research and early clinical trials are encouraging. Multiple research groups have shown that HDAC inhibitors, DNMT inhibitors, and the combination thereof are promising adjuvants that can synergize with and enhance the therapeutic capacity of MAPK inhibitors, immunotherapies, and other therapeutic avenues such as chemotherapy. However, it is critical to emphasize that highly selective epigenetic drugs have less cytotoxic effects than their broad-spectrum counterparts and thus can be used to modulate key cellular processes and antitumoral immune cell functions. Thus, highly specific epigenetic drugs represent a powerful therapeutic approach for those patients that do not benefit from monotherapies currently used in the clinic. 

## Figures and Tables

**Figure 1 cancers-13-06180-f001:**
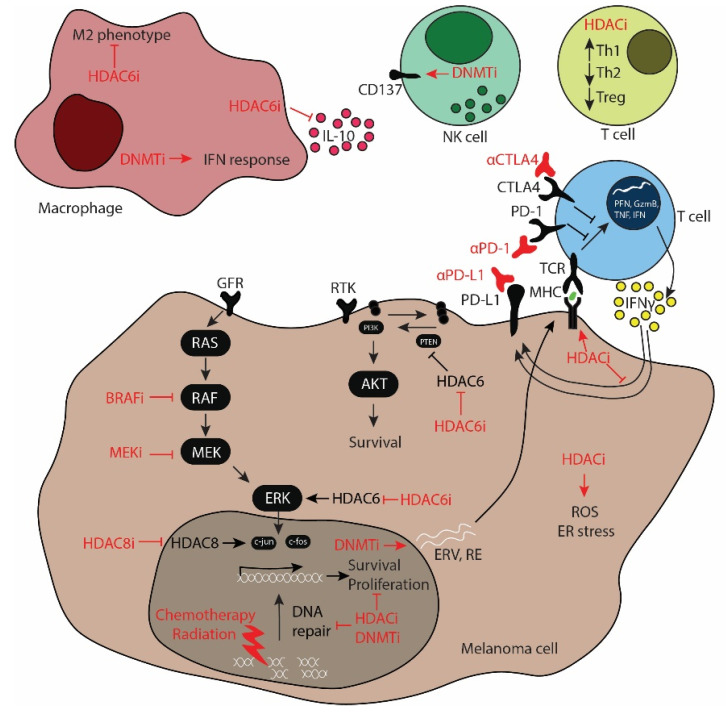
Modulatory effects of epigenetic modifiers in cellular (intrinsic) and immunological (extrinsic) processes. The modulatory effects of HDAC and DNMT inhibitors (HDACi and DNMTi, respectively) in extrinsic factors include the regulation of Th1, Th2, and Treg functions, NK (natural killer) cell activation through CD137, the protumoral M2 macrophage phenotype, secretion of anti-inflammatory cytokines such as interleukin-10 (IL-10), and the stimulation of the interferon (IFN) response. Epigenetic drugs modulate key cellular pathways in melanoma cells. The MAPK pathway activated by GFR (growth factor receptor) signaling activates RAS, RAF, MEK, ERK, c-jun, and c-fos, which regulate the expression of survival and proliferation genes. Similarly, the PI3K/AKT pathway activated by RTKs (receptor tyrosine kinases) leads to the activation of PI3K (PI3 kinase), which then activates AKT and promotes survival. HDACi and DNMTi also modulate the immunogenicity of melanoma cells via increasing antigen presentation in MHC (major histocompatibility complex) and increasing the expression of ERVs (endogenous retroviruses) and RE (repetitive elements), which activates the T cells. Upon activation through TCR (T cell receptor) signaling, T cells release TNF (tumor necrosis factor), GzmB (granzyme B), PFN (perforin), and IFN. HDACi regulate the expression of immunosuppressive genes such as PD-L1 (programmed cell death ligand 1), which can be blocked using anti-PD-1 (programmed cell death 1) antibodies. HDACi generate ROS (reactive oxygen species) and induce ER (endoplasmic reticulum) stress. Additionally, HDACi and DNMTi synergize with radiation therapy and chemotherapy via inhibition of the DNA repair machinery, eventually leading to cell death.

**Table 1 cancers-13-06180-t001:** Epigenetic modifiers regulate the tumor suppressor genes (TSGs) and oncogenes important for melanoma progression.

Gene	TSG or Oncogene	Regulation Level	Epigenetic Regulation	Reference
CDKN2A	TSG	Chromatin	Promoter hypermethylation	[28,29]
MTAP	TSG	Chromatin	Promoter hypermethylation	[30,31]
PTEN	TSG	Chromatin	Promoter hypermethylation	[32,33,34,35]
		Protein	Deacetylation by HDAC6	
RASSF1A	TSG	Chromatin	Promoter methylation	[36,37]
APAF-1	TSG	Chromatin	Promoter methylation	[38]
TP53	TSG	Protein	Deacetylation by HDAC6	[39]
RAS	Oncogene	Protein	Deacetylation by HDAC6 and SIRT2	[40]
ERK	Oncogene	Protein	Acetylation by the CREB-binding protein and p300 Deacetylation by HDAC6	[41]
C-JUN	Oncogene	Protein	Acetylation by p300/CBP Deacetylation by HDAC6 and HDAC8	[42,43]
MITF	Oncogene	Protein Chromatin	Acetylation by CBP/p300 Promoter methylation	[44,45,46]
MDM2	Oncogene	Protein	Deacetylation by HDAC1	[47]
BCL-2	Oncogene	Chromatin	CBP and HDAC2	[48]

**Table 2 cancers-13-06180-t002:** Specificity of epigenetic drugs discussed throughout this review article.

Inhibitor	Target
AR42	All HDACs
Trichostatin A (TSA)	All HDACs
Valproic acid	HDAC1
Panobinostat	All HDACs
Vorinostat (SAHA)	All HDACs
Mocetinostat	HDAC1, HDAC2, HDAC3, HDAC11
ACY-1215	HDAC1, HDAC2, HDAC3, HDAC6, HDAC8
Entinostat	HDAC1, HDAC3
Romidepsin	HDAC1, HDAC2
Tubastatin	HDAC6
PCI-34051	HDAC8
ACY-241	HDAC1, HDAC2, HDAC3, HDAC6, HDAC8
Nexturastat A	HDAC6
Suprastat	HDAC6
Decitabine (5-aza-2′-deoxycytidine)	DNMTs
Guadecitabine	DNMTs

HDAC (histone deacetylase); SAHA (suberoylanilide hydroxamic acid); DNMT (DNA methyltransferase). This table was adapted from Banik et al. [18].

**Table 3 cancers-13-06180-t003:** List of clinical trials in melanoma combining the current therapies with epigenetic modifiers.

Disease	Phase	Therapy	Status	NCT ID
Metastatic melanoma	I	Panobinostat, temozolomide, and decitabine	Safe combination To be continued in a phase II trial	NCT00925132
Stage III and IV melanoma	I	ACY-241 with ipilimumab and nivolumab	Completed	NCT02935790
Metastatic melanoma	II	Entinostat	Completed	NCT00185302
Metastatic uveal melanoma	II	Entinostat with pembrolizumab	Active, not recruiting	NCT02697630
Metastatic melanoma	I	Panobinostat	Completed	NCT01065467
Metastatic melanoma	II	Vorinostat	Side effects, early responses, partial response in two patients and stable disease in most [156]	NCT00121225
Stage III/IV melanoma	I	Panobinostat in combination with ipilimumab	Most patients had serious adverse events	NCT02032810
Malignant melanoma	I	Tinostamustine	Recruiting	NCT03903458
Stage III/IV melanoma	Ib	Mocetinostat in combination with ipilimumab and nivolumab	Terminated	NCT03565406
Metastatic uveal melanoma	II	Entinostat in combination with pembrolizumab	Manageable toxicity, tumor regression, and durable responses	NCT02697630
Stage III/IV melanoma	I	Decitabine	Completed	NCT00002980
Stage III/IV melanoma	I	Decitabine	Completed	NCT00030615
Metastatic melanoma	II	Azacitidine in combination with pembrolizumab	Recruiting	NCT02816021

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
