# Peer review of "Enhancing Therapeutic Approaches for Melanoma Patients Targeting Epigenetic Modifiers"

_cancers, 2021, doi:10.3390/cancers13246180_

Round 1

Reviewer 1 Report

In the manuscript untilled “Enhancing therapeutic approaches for melanoma patients targeting epigenetic modifiers”, the authors performed an interesting review of the various genetic alterations implicated in melanoma development and therapeutic escape. They also suggest alternative therapeutics that may help to circumvent resistance. The manuscript is well documented, written and pleasant to read. A summary scheme would be of great interest and would been support the reading. I recommend realizing this scheme as a minor modification.

Author Response

Thank you for your comments and helpful suggestions. We have included a summary figure (Figure 1) to complement the text.

Reviewer 2 Report

In this manuscript entitled “enhancing therapeutic approaches for melanoma patients targeting epigenetic modifiers”, Gracia-Hernandez summarized melanomagensis focusing on epigenetic modification on the tumor suppressor genes and oncogenes that contribute to melanoma progression and resistance to targeted therapies and immunotherapy. The authors also described the extrinsic mechanism for resistance including the tumor microenvironment, i.e., T cells, macrophage etc. The authors further detailed the current targeted and immune therapies and the combination of these therapies with epigenetic modification drugs. Overall, it is an outstanding and very extensive review of the field. It may even be too dense for the readers to digest. I have the following comments and suggestion:

  1. It may help readers understand by generating cartoons on how the tumor cells (with tumor suppressor gene and oncogenes) and immune cells in the tumor microenvironment respond to the immune therapy and how the epigenetic modifiers may enhance the therapies.
  2. Can the authors elaborate more or generate a table on the current HDAC and DNMT inhibitors and their validated targets? Is there any inhibitor that can modulate immune cell function in tumor microenvironment and meanwhile also activate tumor suppressor gene? The authors indicated the toxicity associated with pan inhibitors. Would the authors suggest combination of different selective inhibitors?
  3. Although the article is focused on melanoma, can the authors have a brief summary whether the mentioned inhibitors in clinical studies have been successfully used in other kinds of tumors?

Author Response

Thank you for the insightful comments and helpful suggestions to improve our manuscript.

We have generated a summary figure (Figure 1) that includes tumor cells and immune cells regarding therapy for melanoma patients. This cartoon differentiates cancer intrinsic and extrinsic mechanisms involving epigenetic modifiers. We have also included a table (Table 2) that lists the epigenetic drugs discussed in this review article and their targets. Additionally, we have added some information on studies evaluating the combination of different epigenetic drugs in section 4. Furthermore, we have included a summary of epigenetic drugs that have been evaluated in clinical studies and are being used to treat hematological malignancies in section 5.

Reviewer 3 Report

Really nice and complete summary of melanoma (combination) therapy.

I have no comments on the content itself. The manuscript is well written and provides an exclllent overview of current therapeutic options.

However, it would benifit greatly form an overview figure of the biological pathways involved in melanoma, including the locations in these pathways (e.g. genes or enzymes)  / where the different therapies have their impact 

Elaborate a bit further (maybe using the suggested figure) why specific combinations would enhance the therapetic effect

Author Response

Thank you for your comments and helpful suggestions. We have included a summary figure and added more information on the mechanism by which epigenetic modifiers can enhance different therapies throughout section 5.

Round 2

Reviewer 3 Report

Nice Figure, I have no further comments